# Study of the Relationship between Pulmonary Artery Pressure and Heart Valve Vibration Sound Based on Mock Loop

**DOI:** 10.3390/bioengineering10080985

**Published:** 2023-08-20

**Authors:** Jiachen Mi, Zehang Zhao, Hongkai Wang, Hong Tang

**Affiliations:** 1School of Biomedical Engineering, Faculty of Medicine, Dalian University of Technology, Dalian 116024, China; mijiachen@mail.dlut.edu.cn (J.M.); zhaozehang@mail.dlut.edu.cn (Z.Z.); wang.hongkai@dlut.edu.cn (H.W.); 2INTESIM (Dalian) Co., Ltd., Dalian 116024, China; 3Liaoning Key Lab of Integrated Circuit and Biomedical Electronic System, Dalian University of Technology, Dalian 116024, China

**Keywords:** pulmonary artery pressure, heart valve vibration sound, mock loop circulation, pulmonary hypertension, backward propagation neural network

## Abstract

The vibration of the heart valves’ closure is an important component of the heart sound and contains important information about the mechanical activity of a heart. Stenosis of the distal pulmonary artery can lead to pulmonary hypertension (PH). Therefore, in this paper, the relationship between the vibration sound of heart valves and the pulmonary artery blood pressure was investigated to contribute to the noninvasive detection of PH. In this paper, a lumped parameter circuit platform of pulmonary circulation was first set to guide the establishment of a mock loop of circulation. By adjusting the distal vascular resistance of the pulmonary artery, six different pulmonary arterial pressure states were achieved. In the experiment, pulmonary artery blood pressure, right ventricular blood pressure, and the vibration sound of the pulmonary valve and tricuspid valve were measured synchronously. Features of the time domain and frequency domain of two valves’ vibration sound were extracted. By conducting a significance analysis of the inter-group features, it was found that the amplitude, energy and frequency features of vibration sounds changed significantly. Finally, the continuously varied pulmonary arterial blood pressure and valves’ vibration sound were obtained by continuously adjusting the resistance of the distal pulmonary artery. A backward propagation neural network and deep learning model were used, respectively, to estimate the features of pulmonary arterial blood pressure, pulmonary artery systolic blood pressure, the maximum rising rate of pulmonary artery blood pressure and the maximum falling rate of pulmonary artery blood pressure by the vibration sound of the pulmonary and tricuspid valves. The results showed that the pulmonary artery pressure parameters can be well estimated by valve vibration sounds.

## 1. Introduction

Pulmonary hypertension (PH) is a hemodynamic disease which is defined as a mean pulmonary arterial pressure (mPAP) of >25 mmHg during right heart catheterization at sea level [1]. Because the etiology of PH is complex and the early clinical features are not obvious, patients with PH have poor survival rates [2]. With the improvement of medical treatment, the incidence and mortality of PH have decreased, but diagnosis is still too late [3]. Invasive right heart catheterization is commonly used to diagnose this disease as the golden standard, but it is a costly approach both in terms of health and finance. So, it is necessary to find a better way to diagnose the disease.

PH is an abnormal hemodynamic state, while the heart sound (HS) is also produced by the movement of blood flow which can contain hemodynamic information; thus, many scholars have shown an interest in the relationship between HS and PH. In 2014, Mohamed et al. [4] collected HS signals from 27 child subjects and conducted a frequency domain analysis. It was found that the energy of mPAP > 25 mmHg was significantly lower than that of mPAP < 25 mmHg in the frequency range of 21 to 22 Hz. Williamet et al. [5] studied the relationship between the acoustic features of the first and second heart sounds (S1 and S2) and hemodynamics in patients with PH based on controlled experiments and found that the acoustic signatures of S1 and S2 were correlated with the severity of pulmonary hypertension and associated with right ventricular dilation and systolic dysfunction. At the same time, the interrelationship between S2 and S1 complexity may also reflect the underlying ventricular interaction of PAH. In 2018, Tang et al. [6] collected the HS signals from 104 subjects, extracted 16 features, and used three entropy algorithms to calculate the heart sound feature sequence to identify PH patients. Wang et al. [7] found that the entropy and energy of HS were strongly related to right ventricular blood pressure (RVBP) in their experiments with beagle dogs.

Many factors can lead to PH. The World Health Organization divides the causes into five categories [8], which makes it hard to control variables in animal experiments. The mock loop (MCL) platform is a good method to avoid the above problems. It does not require ethical approval and is more cost-effective than animal studies [9]. The MCL platform has mainly been used as a test device for ventricular assist devices (VAD). In 1959, Kolff et al. [10] built the first MCL platform. Since then, researchers have tried a variety of designs. In 2005, Timms D et al. [11] proposed a completed MCL, which had two ventricles and open atriums. They used an airtight container to express blood vessel compliance, regulated ball valves to control the resistance of the cardiovascular system and used a check valve instead of heart valves so that experiments on it made good progress. In 2013, Gregor et al. [12] proposed an interface connecting numerical and physical models, which could control MCL by the numerical computer simulation. In 2022, Adji et al. [13] studied the impact of arterial compliance on pump performance and developed arterial pressure. In the same year, Zhao et al. [14] built an MCL platform to investigate the aortic valve performance in patients who used left ventricular assist devices.

In this paper, a circuit platform was built to guide the establishment and regulation of an MCL platform. Then, the relationship between heart valve vibration sound and pulmonary artery blood pressure (PBP) was investigated on the MCL platform by using data acquisition.

## 2. Materials and Methods

The experiments were conducted on the computer model and the MCL platform. The data acquisition device was PL3508 Powerlab8/35 AD Instrument Australia. The reusable blood pressure transducers MLT0380 were used for pressure measurement. For the vibration sound measurement, a MEMS-based heart sound sensor [15] developed by the authors’ lab was chosen, which has the characteristics of high safety and small size.

### 2.1. Circuit Platform

For building and regulating MCL with quantitative guidance, a computer circuit platform was built. The well-known Windkessel model was used in the computer cardiovascular system [16,17,18]. The blood pressure and blood flow are equivalent to the voltage and charge flow. The resistance of blood flow is equivalent to the electronic resistance. The inertia of blood flow can be modeled by the inductance. Inflow and outflow blood to vessel are similar to charging and discharging to linear or nonlinear capacitance. Blood pumping of a heart chamber can be simulated by a nonlinear voltage source with respect to volume and time. Valves in the heart and vessels are like diodes. Therefore, a circuit model for the human pulmonary system is proposed in this study and taken as a platform to simulate pulmonary hemodynamics. The pressure–volume (P-V) relation of a segment of vessel is generally modeled by a two-element Windkessel model: resistance and compliance. The circuit platform of pulmonary circulation is shown in the Appendix A. Blood is pumped out of the right ventricle (RV), passing through the pulmonary valve (*D_p_*) and entering the pulmonary artery. Blood flows through pulmonary vessels which are simulated by three parts: proximal pulmonary artery (pap), distal pulmonary artery (pad) and pulmonary veins (pv). Finally, blood pressure enters the left atrium (LA).

The key point of this paper is pulmonary circulation, so the authors built the lumped parameter circuit platform for the pulmonary circulation system only, according to the systemic circulation part and the heart part used in the work of Tang et al. [19]. The pulmonary circulation based on the Windkessel model consists of *p*-valve, pap, pad and pv, and every part corresponds to a part of the MCL platform discussed in the Section 2.2. The relations between compliance *C*, inductance *L*, blood flow *Q* and blood pressure *P* in the circuit system are:(1)$$Q(t)=C\frac{dP}{\mathrm{dt}},\;P(t)=L\frac{dQ}{\mathrm{dt}}.$$
solving differential equations, they can be written as:(2)$$V^{\prime }\left(t\right)=\frac{dV(t)}{\mathrm{dt}}=Q(t),\;Q^{\prime }\left(t\right)=\frac{dQ(t)}{\mathrm{dt}}=\frac{P(t)}{L},$$

Then, the normal state of the circulation system can be obtained. The parameters and initial values of the elements in the model are given in Appendix B. The pressure–volume (P-V) loop of the right ventricle is shown in Figure 1. The blood pressure and blood flow at some pulmonary nodes are shown in Figure 2a,b. It can be seen that the P-V loop that the platform generated is similar to that of a normal person. The systolic pressure in the right ventricle is about 23 mmHg, and the isovolumic state is clear in Figure 1. Figure 2a,b shows that the simulated pulmonary circulation system works in a normal state.

To achieve pulmonary hypertension status, we increased the resistance of pad, which is also the simplest and most effective method. The blood flow resistance, *R*, is inversely proportional to the fourth power of *r*, which is the radius of the pipe,
(3)$${R=8\eta L/\pi r}^{4}$$
where *L* is the pipe length and *η* is the liquid viscosity. Assume *L* and *η* are constants. In order to simulate the development of distal pulmonary arteries narrowing over time, the radius *r* of decreases as a function of time, that is
(4)$${r(t)=r}_{0}{{(1+g}_{r}\;\times \;t)}^{-1/4}$$
where $r_{0}$ is the initial radius and $g_{r}$ is the changing rate. This condition is also called distal pulmonary artery stenosis (DPAS). Typical P-V loops of the right ventricle caused by DPAS are shown in Figure 3, where RVBP is increasing over time because it takes more pressure to pump the blood out of the right ventricle.

### 2.2. Mock Loop Platform

The MCL platform we designed has the following characteristics: (1) It has the same pulmonary circulation structure as the computer model. (2) Its distal pulmonary artery resistance can be adjusted continuously. (3) It can produce valve vibration sounds similar to that of the human body. On the basis of learning from previous research achievements [20,21,22,23,24], the concept diagram of the MCL platform is shown in Appendix A. In systemic and pulmonary circulation, when the servo motors move downwards, the left and right ventricles are squeezed and pump liquid out, while the aortic valve and pulmonary artery valve open. After passing through the compliance device and resistance charge, blood flows into the reservoirs. When the servo motors move upwards, the tricuspid valve and mitral valve open, and blood flows back into the ventricle. An image of the MCL platform is shown in Appendix A.

There is an equivalent relationship between the mock loop model and the circuit model. By comparing Appendix A, it can be observed that the corresponding relationship is as follows. Appendix A corresponds to the right ventricle (RV), Appendix A corresponds to the pulmonary valve (*p*-valve), the tube path between Appendix A corresponds to the proximal pulmonary artery (pap), the tube path between Appendix A corresponds to the distal pulmonary artery (pad), and Appendix A corresponds to the pulmonary vein (pv).

### 2.3. Ventricle

In order to better reflect the direct interaction between the left and right ventricles, we designed a biventricular model which has left and right ventricles and a ventricular septum, as shown in Appendix A. The ventricle model is composed of silica gel material with a Shore hardness of 70 degrees, and the septum is covered with hard, thin plastic to prevent the left heart from pressing heavily on the right heart.

### 2.4. Drive and Fixed Part

In order to squeeze the ventricle evenly, we chose the hydraulic method instead of applying the motor directly to the ventricular model. The drive and fixed part are shown in Appendix A. The whole unit is filled with water, and the servo motor simulates ventricular ejection by driving water to squeeze the ventricles evenly.

### 2.5. Compliance

According to the three-element Windkessel model, there are three characteristics in the vascular system: resistance, inductance and compliance. Compliance refers to the deformation ability of an elastomer under the external forces, which plays a cushioning and regulating role in the vascular system. Because there is a difference in the value of compliance between systemic and pulmonary circulation, a compliance device with adjustable parameters was developed.

The schematic diagram of the compliance device we proposed is shown in Appendix A, which is designed based on air spring. We can control the volume of sealed gas by adjusting the position of the piston, thus controlling the compliance. According to the ideal gas equation and the definition of compliance, the compliance can be calculated as:(5)$$C=\frac{{(h}_{\mathrm{fluid}}\cdot {(h}_{\mathrm{can}}{\;-\;h}_{\mathrm{fluid}})\cdot A_{\mathrm{can}})}{P_{\mathrm{atom}}\cdot h_{\mathrm{can}}+{(\rho gh}_{\mathrm{fluid}}{\;-\;P}_{\mathrm{atom}})\cdot {(h}_{\mathrm{can}}-h_{\mathrm{fluid}})}$$

In (5), *C* is the compliance value of the device, $h_{\mathrm{fluid}}$ is the height of fluid, $h_{\mathrm{can}}$ and $A_{\mathrm{can}}$ are the height and bottom area of the devices, $P_{\mathrm{atom}}$ is the pressure of atmosphere, and ρg is the product of the liquid density and the local gravity.

### 2.6. Reservoir

The reservoir is a container connected to the atmosphere, which simulates the veins and atria in systemic and pulmonary circulation. It relies on gravity to simulate the compliance of the veins and atria.

### 2.7. Valve and Resistance

In order to make the vibrational sound more similar to that of the real heart valve, we used four artificial mechanical heart valves, which were produced by LANFEI MEDICAL in Lanzhou China. The heart valve is composed of a single blade attached to the fixed part.

A laboratory throttle valve was used to adjust resistance, and the cross-sectional area of the vessel can be changed by adjusting the screw on it.

### 2.8. Servo Motor

Two stepper motors, a control box and two trigger switches were chosen as the servo motor system of the MCL platform. After the motor reached the reference position by triggering the switch, the linear reciprocating motion was carried out according to the program, which is determined by the circuit platform.

## 3. Experiment Procedure

After the above device was installed, the ventricular ejection program should be written into the servo motor. Because it is a flow control mode, the corresponding aorta and pulmonary artery flow should be referenced to determine the program. After the experiments and considering the limited motor power and experimental conditions, some corrections must be made to the program. Let the blood flow out for a longer period of time while the stroke volume of the two ventricles is 67 mL.

After writing the program into the servo motor, fill the fixtures with water and add appropriate liquid, which consists of glycerin and water, to the vascular system. Then, the pressure waves of the right ventricle and pulmonary artery can be measured, as shown in Figure 4.

In Figure 4, two points need to be explained. First, the systolic pressure of the right ventricle was about 50 mmHg, and the systolic pressure of the pulmonary artery was 28 mmHg, both of which are higher than those in a normal human. This was due to limited experimental conditions, such as the single-bladed heart valve, which is again different than in a normal human. Because we were mainly looking for rules in the changes of blood pressures, this problem could be ignored. Second, since the return of blood to the ventricle depends on fluid gravity in the reservoir rather than by the servo motor, the diastole phase cannot be controlled. That is why diastolic blood pressure was not chosen as the research object.

By adjusting the radius of the distal pulmonary artery, pulmonary artery systolic blood pressure (PASP) can be controlled to different values. In this paper, six pulmonary hypertension states were simulated, which meant six different pressure parameters of the pulmonary artery. Each experiment lasted 5 min, and pulmonary pressure, pulmonary valve and tricuspid valve vibration were collected synchronously. The pulmonary artery pressure signal information obtained in the experiment is shown in Table 1. As can be seen, the signals obtained in each experiment were relatively stable. The typical pulmonary artery pressure signals obtained in the six experiments are shown in Figure 5. It was found that as the resistance of the distal pulmonary artery increased, the baseline pulmonary artery pressure increased, corresponding to human physiological conditions.

## 4. Data Process and Analysis

The pressure signal was filtered by using a Butterworth low-pass filter, and the vibration signal was denoised using the wavelet threshold method. The servo motor pulse signal represented the direction of motor motion, by which systolic and diastolic periods can be divided, that is, a high level represented the systole and a low level represented the diastole. To some extent, we can think of this as the role of the ECG signal in cycle segmentation. After denoising the servo motor signal, the rising and falling edges of the signal were extracted. The maximum blood pressure during the high level of the servo motor signal was taken as the systolic blood pressure of PBP, the maximum rising rate of pulmonary artery blood pressure was defined as MRRPBP, and the maximum falling rate of pulmonary artery blood pressure was defined as MFRPBP. Based on time windows and servo motor pulse signals, the time segments containing the tricuspid and pulmonary valve vibration sounds can be segmented. In the time window, the position of the maximum signal amplitude was set as the center and extracted 98% of the energy concentration area as the corresponding vibration sound of the valve.

Several characteristics of vibration sounds with physical significance were extracted and studied. Maximum amplitude (*Amp*) was defined as the maximum value of vibration sound, and the timing of the maximum amplitude (*MS*) was the time interval from the falling edge of the motor waveform to maximum amplitude occurrence. Sample entropy (*SampEn*) and fuzzy entropy (*FuzzyEn*) were also in the features, of which the dimensions were both 2; the threshold values were both 0.2 times the standard deviation, and the step of *FuzzyEn* was 2. The power spectral density of the vibration sound was calculated based on the burg algorithm, and the amplitude (*Pow*) and frequency (*f*) of the power spectral density main peak were also extracted. The letters ‘*t*’ and ‘*p*’ after the underline of the feature name denoted the tricuspid and pulmonary valves. For example, *Amp_t* represents the maximum amplitude of the tricuspid vibration sound. More detailed information of the features is shown in Table 2.

## 5. Statistical Method

Various features are proposed in this study. In order to check the performance of the features to link the pressure parameters (PBP, MRRPBP and MFRPBP), a statistical method was necessary. It is difficult to study the significance between these features and pressure parameters without knowing the features’ distribution patterns. The authors used the Kruskal–Wallis rank sum test. This is a nonparametric test with the goal of determining if all group samples are identical or if at least one of the groups tends to give observations that are different from those of other populations.

## 6. Results

The Kruskal–Wallis rank sum test was used to evaluate whether there were significant differences between the groups of each feature. Figure 6 and Figure 7 show the difference comparison results and features’ boxplot of the tricuspid valve vibration sound between groups. In Figure 6, dark blue represents a significant difference between two groups, yellow indicates rejection of the hypothesis of significant difference between the groups, and green denotes self-comparison within the group, which has no practical significance.

Combined with Figure 6 and Figure 7, the changes of each feature with the increased resistance of the distal pulmonary artery could be observed. In several features, *MS_t* had significant differences between each group and gradually decreased with the increase in resistance. This was because the slope of right ventricular pressure increased while the right ventricular EDPVR remained unchanged, which made the right ventricular pressure greater than the pressure of the body circulation reservoir earlier, resulting in early tricuspid valve closure and smaller *MS_t*. In addition, the amplitude characteristics of tricuspid valve vibration sound *Amp_t* and energy characteristics *Ener_t* tended to increase, which was because the ventricle needed more pressure to eject liquid, resulting in an increased pressure difference between the two sides of the valve. In the frequency characteristics, *f_t* decreased while *Pow_t* increased, indicating that the main peak frequency of the tricuspid valve vibration sound power spectrum decreased and its amplitude increased with the increase in peripheral resistance in this experiment. Among the changes in frequency characteristics, the change trend of the main peak frequency in this experiment was contrary to the experience of the auscultation of heart sounds and previous research results. The reason may be that the heart valves used in this paper comprise mechanical valves composed of stainless steel and carbon materials, leading to the change trend contrary to biological valves. However, the work in this paper can still reflect the changes in the frequency of vibration sounds.

The difference comparison results and features’ boxplot of pulmonary valve vibration sounds between groups are shown in Figure 8 and Figure 9. In the analysis of the vibration sound features of the pulmonary valve, *MS_p* and *Ener_p* had significant differences between each group, and *MS_p* decreased gradually, which was consistent with the study conducted by Wang et al. [7] on the second heart sound. In addition, *Ener_p* and *AMP_p* increased gradually, which was because the compliance of pulmonary circulation decreased with the increase in peripheral resistance, resulting in the increase in blood pressure in the pulmonary artery. When the right ventricle entered the diastole period, the pulmonary artery blood accelerated the reflux, which made the pulmonary valve close with greater energy. In the frequency characteristics, *f_p* decreased while *Pow_p* increased, indicating that the main peak frequency of the vibration sound power spectrum of the pulmonary valve decreased and its amplitude increased with the increase in peripheral resistance.

## 7. Prediction of Blood Pressure Parameters by Valve Vibration Sound

In the above analysis, the relationship between vibration sound and pulmonary arterial pressure was verified. We suspected whether the pressure parameters could be predicted by the valve vibration sound. To prove this idea, an experiment was designed. We obtained continuous changes in pulmonary artery pressure by adjusting the distal vascular radius of the pulmonary artery. The experiments showed that as the radius of the distal pulmonary artery decreased, the flow resistance gradually increased, and the pressure of the pulmonary artery gradually increased, and vice versa. The vibration sounds of the tricuspid and pulmonary valves were collected simultaneously. Each experiment lasted 10 min, with each carried out five times in total. Table 3 shows the details of the experimental data.

First, a backward propagation neural network was used in the fitting. In these experiments, seven valve vibration sound features were extracted, which were highly correlated with the three pulmonary artery pressure parameters, i.e., PASP, MRRPBP and MFRPBP. These parameters may be different from the group analysis above because continuous adjustment parameters may cause the system to always be in an unbalanced state. These seven features and their correlation coefficients (CC) with pressure parameters are shown in Table 4. From the analysis of correlation, it can be seen that some features had a relatively high correlation with the pressure parameters. For example, the peak timing of pulmonary valve vibration waveform correlated with PASP, MRRPP and MFRPP up to −0.94, −0.90 and 0.80. Therefore, those features with high correlation could be used to predict the pressure parameters.

A backward propagation neural network, a traditional regression prediction tool, was used here to predict three pulmonary artery pressure parameters. The number of input layer cells was 7 (seven features listed in Table 4), the number of hidden layers was 2, the number of hidden layer cells was 10 and the number of output units was 3 (three pressure parameters shown in Table 4). The structure diagram is shown in Appendix A.

In this experiment, the inter-group verification method was adopted, that is, four groups of data were used as the training set, and the other group of data was used as the verification set. The scatter plot of predicted and measured values is shown in Figure 10. The mean absolute error (MAE) of regression estimations of the three parameters were 1.187 mmHg, 0.0023 mmHg·ms^−1^ and 0.0042 mmHg·ms^−1^, respectively. The result showed that the pressure parameters could be estimated well.

Machine learning does not seem to be automated enough, and we wish to solve this problem by deep learning. Therefore, a deep learning model constructed by a CNN and Bi-LSTM network was proposed in this paper. In this model, CNN was used for feature extraction, which can extract the features of a single vibration sound or two vibration sounds jointly. Bi-LSTM was used to learn the time sequence features in the sequence and output the regression results by the full connection layer. The structure diagram of the model is shown in Appendix A. Tricuspid and pulmonary valve vibration sound were input, and three pulmonary pressure features were output.

The inter-group verification method was also adopted in this model. A scatter plot of predicted and measured values is shown in Figure 11. The MAEs of the regression estimation of the three parameters were 3.403 mmHg, 0.0034 mmHg·ms^−1^ and 0.0060 mmHg·ms^−1^, respectively. The performance of the deep learning model was worse than that of the backward propagation neural network model, especially at both ends of the parameter range. The reason for this may be the small number of samples in that range. After removing the samples with PASP > 80 mmHg and PASP < 40 mmHg, the MAEs of the three pressure parameters’ estimation were increased to 3.17 mmHg, 0.0033 mmHg∙ms^−1^ and 0.0058 mmHg∙ms^−1^, respectively.

## 8. Discussions

This study discovered the connection between valve vibration and pressure parameters through the mock loop model, as well as predicted pulmonary pressure parameters based on the features of vibration waveform. Some valve vibration features discovered through the mock loop model in this study were consistent with clinical observations to a certain degree, as shown in Table 4. For example, both the timing of pulmonary and tricuspid valve vibration waveforms reduced with respect to increasing parameters. The amplitude of the vibration sound of the pulmonary valve was positively correlated to pulmonary pressure.

As we all know, the pressure parameters of pulmonary circulation have important value in monitoring lung function and diagnosing pulmonary arterial hypertension. However, the pressure parameters of pulmonary circulation are not routine monitoring items in clinical practice and are invasive. The significance of the research results in this article lies in proposing a possible method to estimate pulmonary circulation blood pressure parameters using heart valve vibrations collected from the chest surface, thereby achieving continuous non-invasive monitoring. If an electronic monitoring instrument is developed based on this method, then low-cost, real-time, fast and convenient monitoring of pulmonary circulation blood pressure parameters could be achieved.

## 9. Study Limitations

The findings of this study must be seen in light of some limitations. It is known that the human pulmonary circulatory system is very complex, and neither the circuit model nor the mock loop model could perfectly simulate it. The valves used in the mock loop model are artificial mechanical valves, whose material and structure are largely different from those of human heart valves. The features of artificial mechanical valve vibration could differ from human heart valves. On the other hand, the changes in its hemodynamics are influenced by many factors. This study only considers a single factor: blood flow resistance caused by the stenosis of distal pulmonary circulation vessels. The rules of valve vibration under multiple factors are still open to be explored.

## 10. Conclusions

In this paper, the relationship between pulmonary artery pressure and heart valve vibration sound was studied based on MCL. It was found that when the pulmonary artery pressure rose, the feature of valve-closing vibration sound changed in both the time and frequency domains, which were mainly reflected in the amplitude, energy and time domains. In addition, the continuously changing pulmonary artery pressure was extracted on the MCL, and the machine learning and deep learning methods were proposed to verify that the pressure-related parameters could be predicted by the valve vibration sounds. This paper provides an in vitro experimental basis for the non-invasive diagnosis of pulmonary hypertension and the non-invasive measurement of pulmonary arterial pressure.

## Figures and Tables

**Figure 1 bioengineering-10-00985-f001:**
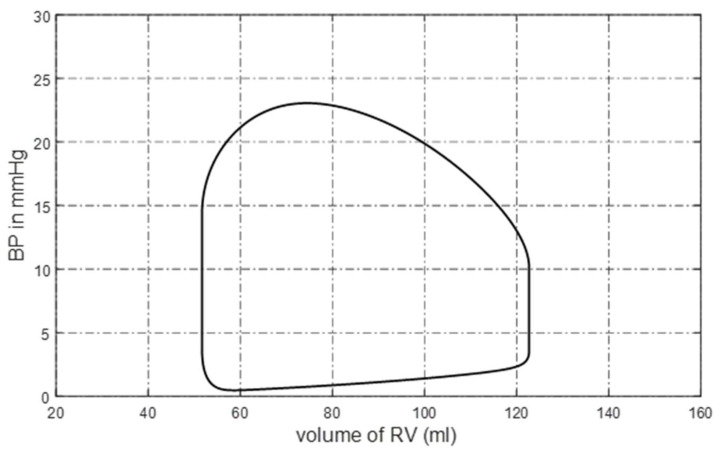
P-V loop of right ventricle.

**Figure 2 bioengineering-10-00985-f002:**
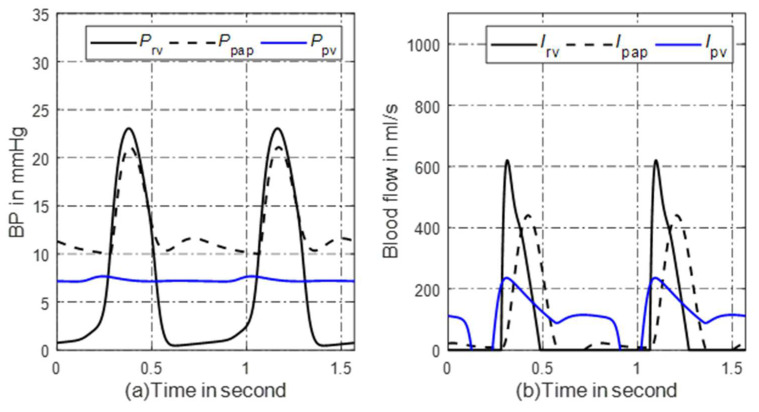
Simulated hemodynamics of pulmonary circulation. (**a**) Key pulmonary blood pressures; (**b**) corresponding blood flows of (**a**). *P*: blood pressure; *I*: blood flow; rv: right ventricle; pap: proximal pulmonary artery; pv: pulmonary veins.

**Figure 3 bioengineering-10-00985-f003:**
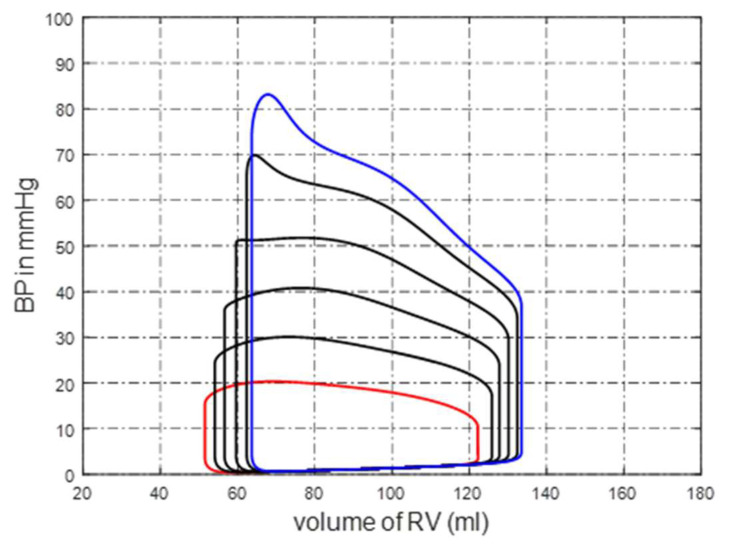
P-V loops of right ventricle caused by DPAS. The different colours lines mean the progress of DPAS. The red and blue colours lines indicate normal and sever DPAS.

**Figure 4 bioengineering-10-00985-f004:**
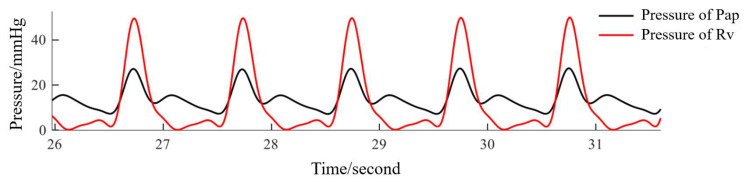
Pressure signal of pulmonary artery and right ventricle.

**Figure 5 bioengineering-10-00985-f005:**
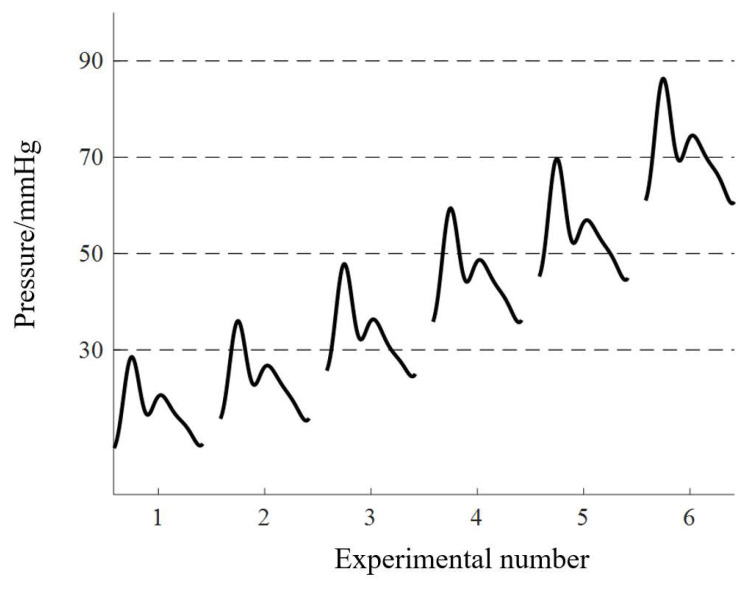
Typical waveform of pulmonary arterial pressure in 6 experiments.

**Figure 6 bioengineering-10-00985-f006:**
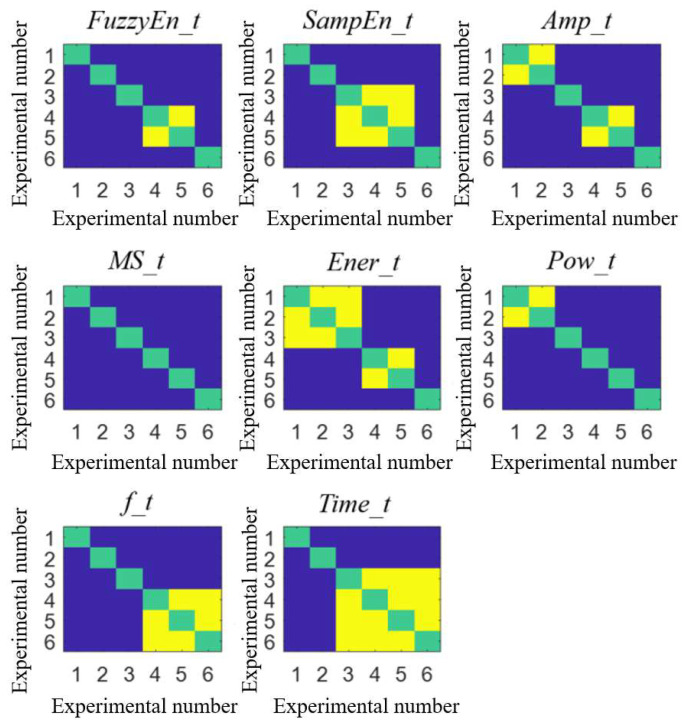
Results of difference in pairs between groups of tricuspid valve vibration. Green color indicates self relationship within the group, dark blue color indicates significant differences between the two groups, and yellow color indicates no significant differences between the two groups.

**Figure 7 bioengineering-10-00985-f007:**
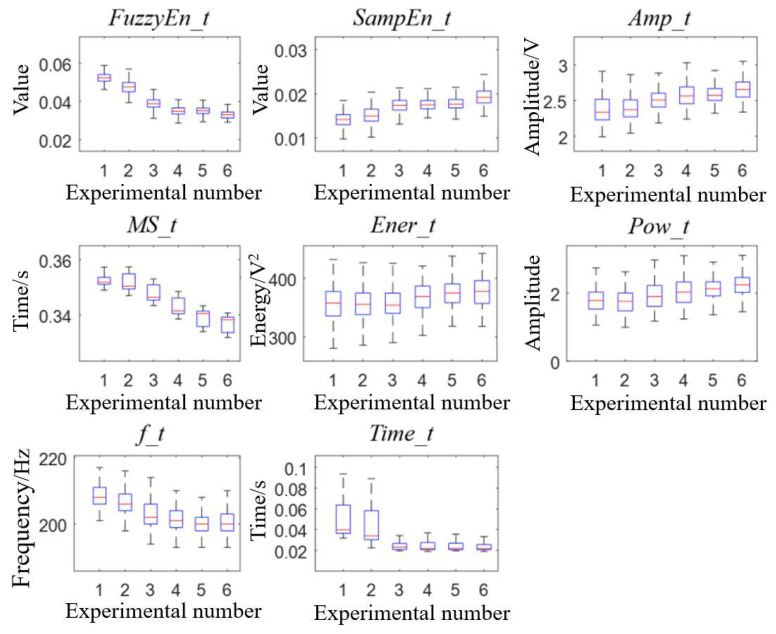
Boxplot of 8 tricuspid valve vibration characteristics in 6 experiments.

**Figure 8 bioengineering-10-00985-f008:**
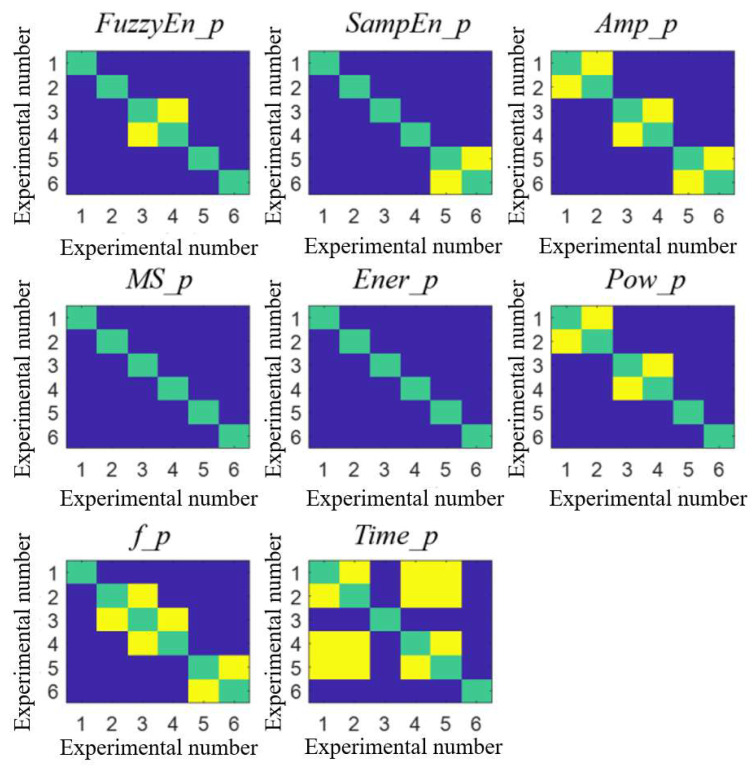
Results of difference in pairs between groups of pulmonary valve vibration. Green color indicates self relationship within the group, dark blue color indicates significant differences between the two groups, and yellow color indicates no significant differences between the two groups.

**Figure 9 bioengineering-10-00985-f009:**
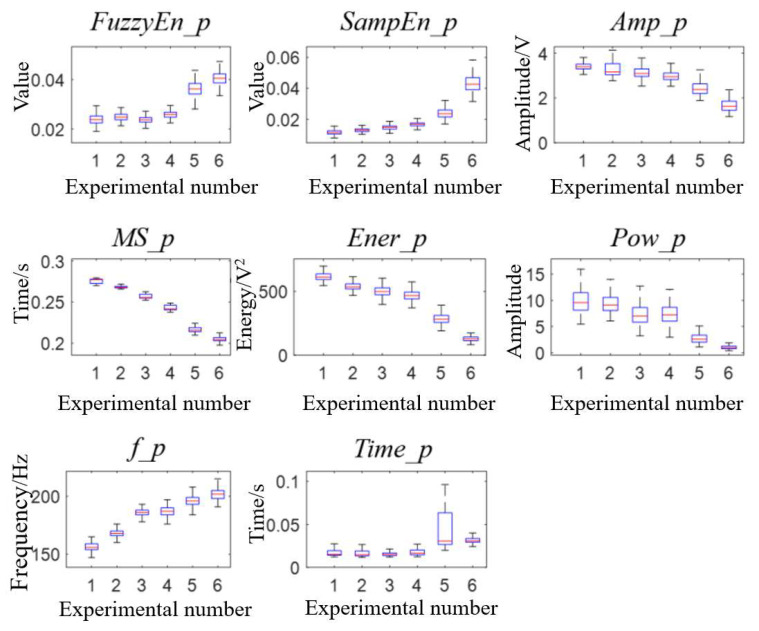
Boxplot of 8 pulmonary valve vibration characteristics in 6 experiments.

**Figure 10 bioengineering-10-00985-f010:**
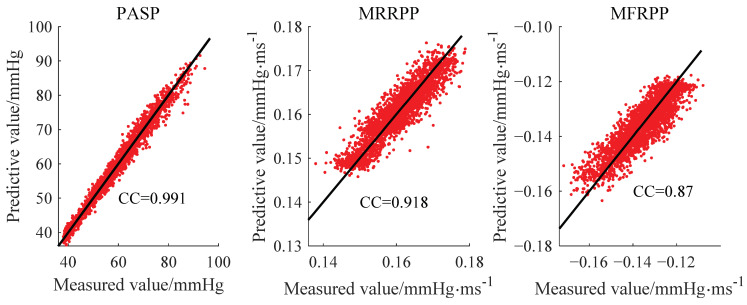
Scatter plot of predicted and measured values.

**Figure 11 bioengineering-10-00985-f011:**
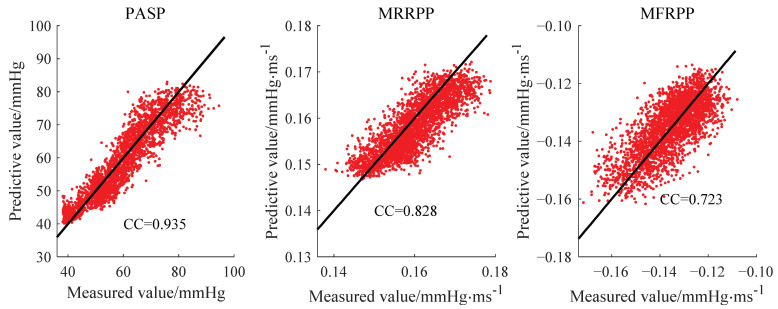
Scatter plot of predicted and measured values.

**Table 1 bioengineering-10-00985-t001:** Pulmonary artery pressure signal information.

Experiment Number	Mean Value ± Standard Deviation		
	PASP/mmHg	MRRPP/mmHg·ms^−1^	MFRPP/mmHg·ms^−1^
1	28.50 ± 0.17	0.14 ± 0.0024	−0.11 ± 0.0033
2	36.01 ± 0.18	0.15 ± 0.0025	−0.12 ± 0.0036
3	47.85 ± 0.19	0.16 ± 0.0028	−0.14 ± 0.0037
4	59.84 ± 0.22	0.18 ± 0.0031	−0.14 ± 0.0041
5	69.72 ± 0.24	0.18 ± 0.0032	−0.15 ± 0.0045
6	86.51 ± 0.51	0.19 ± 0.0034	−0.15 ± 0.0046

Note: PASP, pulmonary artery systolic blood pressure; MRRPP, maximum rising rate of pulmonary artery blood pressure; MFRPP, maximum falling rate of pulmonary artery blood pressure.

**Table 2 bioengineering-10-00985-t002:** Information of features.

Physical Meaning	Name	
	Tricuspid Valve	Pulmonary Valve
Maximum value	*Amp_t*	*Amp_p*
Energy	*Ener_t*	*Ener_p*
Timing of maximum value	*MS_t*	*MS_p*
Time of duration	*Time_t*	*Time_p*
Main peak frequency of power spectrum	*f_t*	*f_p*
Main peak amplitude of power spectrum	*Pow_t*	*Pow_p*
Sample entropy	*SampEn_t*	*SampEn_p*
Fuzzy entropy	*FuzzyEn_t*	*FuzzyEn_p*

**Table 3 bioengineering-10-00985-t003:** Information of pulmonary artery pressure.

Experiment Number	Number of Cardiac Cycles	Range		
		PASP/mmHg	MRRPP/mmHg·ms^−1^	MFRPP/mmHg·ms^−1^
1	581	38.13~94.50	0.14~0.18	−0.17~−0.11
2	581	38.75~89.33	0.14~0.18	−0.17~−0.11
3	584	38.05~91.21	0.14~0.18	−0.17~−0.11
4	584	49.26~85.66	0.15~0.18	−0.17~−0.12
5	584	49.36~81.80	0.15~0.18	−0.17~−0.11

Note: PASP, pulmonary artery systolic blood pressure; MRRPP, maximum rising rate of pulmonary artery blood pressure; MFRPP, maximum falling rate of pulmonary artery blood pressure.

**Table 4 bioengineering-10-00985-t004:** CC values between vibration sound features and pulmonary artery pressure parameters.

Feature	Correlation Coefficient		
	PASP	MRRPP	MFRPP
*MS_p*	−0.94	−0.90	0.80
*Ener_p*	0.54	0.51	−0.48
*Amp_p*	0.31	0.29	−0.27
*f_p*	−0.40	−0.37	0.37
*FuzzyEn_t*	−0.67	−0.67	0.56
*MS_t*	−0.31	−0.21	0.23
*f_t*	−0.26	−0.25	0.20

Note: PASP, pulmonary artery systolic blood pressure; MRRPP, maximum rising rate of pulmonary artery blood pressure; MFRPP, maximum falling rate of pulmonary artery blood pressure.

## Data Availability

The data is available with request.

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
