# Peer review of "Study of the Relationship between Pulmonary Artery Pressure and Heart Valve Vibration Sound Based on Mock Loop"

_bioengineering, 2023, doi:10.3390/bioengineering10080985_

Round 1

Reviewer 1 Report

The authors performed a study to evaluate valve vibration with PHT.

The manuscript is interesting. Some minor comments to improve it:

No need to use unrepeated abbreviations in the abstract 

A separate statistical section in the methods is recommended 

Please explain figure 1.

Discuss section can be merged into one section after the results 

Please Discuss the study limitations 

What is the correlation to the clinical practice? In patients with PHT, second heart sound is accentuated. 

Please provide a take home message or conclusion at the end of the manuscript 

Accepted

Author Response

Dear Editors and Reviewers:

Thank you for your letter and for the reviewers’ comments concerning our manuscript entitled “Study of Relationship Between Pulmonary Artery Pressure and Heart Valve Vibration Sound based on Mock Loop” (Manuscript ID: bioengineering-2548545). Those comments are all valuable and very helpful for revising and improving our paper, as well as the important guiding significance to our researches. We have studied comments carefully and have made revisions which we hope meet with approval. Revised portion are marked in red in the paper. The main corrections in the paper and the responds to the reviewer’s comments are as flowing.

Reviewer 1

The authors performed a study to evaluate valve vibration with PHT. The manuscript is interesting. Some minor comments to improve it:

(1) No need to use unrepeated abbreviations in the abstract

Answer: the unrepeated abbreviations are removed from the abstract, such as HS, MCL, PBP, RVBP, etc.

(2) A separate statistical section in the methods is recommended

Answer: The authors supplemented a statistical subsection to explain the statistical method used in this study. It is shown in the following.

Statistical method

Various features are proposed in this study. In order to check the performance of the features to link the pressure parameters (PBP, MRRPBP and MFRPBP), a statistical method is necessary. It is difficult to study the significance between these features and pressure parameters without knowing the features’ distribution patterns. The authors used the Kruskal–Wallis rank sum test. It is a nonparametric test that has as its goal to determine if all group samples are identical or if at least one of the groups tends to give observations that are different from those of other populations.

(3) Please explain figure 1.

Answer: The authors have explained the principle to build the circuit model for pulmonary circulation system shown in Fig.1. In the revised version, the paragraph to explain Fig. 1 becomes the following.

For building and regulating MCL with quantitative guidance, a computer circuit platform is built firstly. The well-known Windkessel model is used in the computer cardiovascular system [16-18]. The blood pressure and blood flow are equivalent to the voltage and charge flow. The resistance of blood flow is equivalent to the electronic resistance. The inertia of blood flow can be modeled by the inductance. Inflow and outflow blood to vessel are similar to charging and discharging to linear or nonlinear capacitance. Blood pumping of a heart chamber can be simulated by a nonlinear voltage source with respect to volume and time. Valves in heart and vessels are like diodes. Therefore, a circuit model for human pulmonary system is proposed in this study and taken as a platform to simulate pulmonary hemodynamics. The P-V relation of a segment of vessel is generally modeled by a two-element Windkessel: resistance, and compliance. The circuit platform of pulmonary circulation is shown as Fig.1. Blood is pumped out of the right ventricle (RV), passing through the pulmonary valve (Dp) and entering the pulmonary artery. Blood flows through pulmonary vessels which are simulated by three parts: proximal pulmonary artery (pap), distal pulmonary artery (pad), and pulmonary veins (pv). Finally, blood pressure enters the left atrium (LA).

(4) Discuss section can be merged into one section after the results

Answer: The Discussion sections have been merged into one section. Thank you very much!

(5) Please Discuss the study limitations

Answer: thank you for the comment. The authors have indicated study limitations in a separate section of “Study limitations”. The new section is shown in the following.

Study limitations

  The findings of this study have to be seen in light of some limitations. The human pulmonary circulatory system is very complex. Both the circuit model and the mock-loop model cannot perfectly simulate the human pulmonary circulatory system. The valves used in the mock-loop model are artificial mechanical valves, whose material and structure are largely different from those of human heart valves. The features of artificial mechanical valve vibration could be away from human heart valves. On the other hand, the changes in its hemodynamics are influenced by many factors. This study only considers a single factor: blood flow resistance caused by stenosis of distal pulmonary circulation vessels. The rules of valve vibration under multiple factors are still open to be explored.

(6) What is the correlation to the clinical practice? In patients with PHT, second heart sound is accentuated.

Answer: After the authors’ analysis, it is found that there is a certain correlation between the valve vibration characteristics exhibited by the mock-loop model and clinical observation results. For example, as the pulmonary artery blood pressure increase in the mock-loop model, the features of amplitude, energy and power of tricuspid valve increase, as well as the vibration frequency of pulmonary valve increases simultaneously. These results are somewhat close to clinical observation from PHT patients. However, due to technical reasons, it is not yet possible to measure the vibration characteristics of the tricuspid valve and pulmonary artery valve separately on the patient's heart. Therefore, some of the valve vibration patterns discovered in this study have not yet received sufficient clinical validation. We look forward to solving this problem with future technological advancements.

(7) Please provide a take home message or conclusion at the end of the manuscript

Answer: the authors have provided a Conclusion section at the end of manuscript. It is shown in the following.

Conclusions

The relationship between pulmonary artery pressure and heart valve vibration sound was studied based on MCL. It was found that, when the pulmonary artery pressure rises, the feature of valve closing vibration sound changes in both the time and frequency domains, which were mainly reflected in the amplitude, energy, time domain. In addition, the continuously changing pulmonary artery pressure was extracted on the MCL, and the machine learning and deep learning methods were proposed to verify that the pressure related parameters could be predicted by the valve vibration sound. This paper provides an in vitro experimental basis for the non-invasive diagnosis of pulmonary hypertension and the non-invasive measurement of pulmonary arterial pressure.

  By the way, the manuscript has been polished in English by an English teacher. The authors believe that the revised version is easy to read.

Reviewer 2 Report

The manuscript entitled Study of Relationship Between Pulmonary Artery Pressure and Heart Valve Vibration Sound based on Mock Loop is an original article. The authors analyzed the relationship between vibration sound and pulmonary arterial pressure based on MCL. They have registered continuous changes in pulmonary artery pressure by continuously adjusting the vascular radius of the pulmonary artery. They have found that when the pulmonary artery pressure rises, the feature of vibration sound produced by the heart valve closing changes in both the time and frequency domains, mainly reflected in the amplitude, energy, time domain. The machine learning and deep learning methods were used to verify that the pressure parameters could be predicted by the vibration sound of the heart valve. In this experiment, the authors provided a theoretical basis for the non-invasive diagnosis of pulmonary hypertension and the non-invasive measurement of pulmonary arterial pressure by simulation.

The article is interesting because the gold standard for pulmonary artery pressure is right heart catheterization (an invasive method). A non-invasive method for this assessment would be a great discovering.

The article is well written and well structured. The methods are well described. Discussions are actually the conclusions. Please make a real chapter of discussions about the importance of the results in the context of data published until now.

What are the limitations study? Please clearly specify.

Minor revision

There are not legends for tables.

There are too many figures. I suggest putting some figures in supplementary materials (figures 1,5,6, 7, 8, 9, 16, 18.

There are no many references probably because this study theme was not frequently searched.

There are some minor  English language errors.

Author Response

Dear Editors and Reviewers:

Thank you for your letter and for the reviewers’ comments concerning our manuscript entitled “Study of Relationship Between Pulmonary Artery Pressure and Heart Valve Vibration Sound based on Mock Loop” (Manuscript ID: bioengineering-2548545). Those comments are all valuable and very helpful for revising and improving our paper, as well as the important guiding significance to our researches. We have studied comments carefully and have made revisions which we hope meet with approval. Revised portion are marked in red in the paper. The main corrections in the paper and the responds to the reviewer’s comments are as flowing.

Reviewer 2

The manuscript entitled Study of Relationship Between Pulmonary Artery Pressure and Heart Valve Vibration Sound based on Mock Loop is an original article. The authors analyzed the relationship between vibration sound and pulmonary arterial pressure based on MCL. They have registered continuous changes in pulmonary artery pressure by continuously adjusting the vascular radius of the pulmonary artery. They have found that when the pulmonary artery pressure rises, the feature of vibration sound produced by the heart valve closing changes in both the time and frequency domains, mainly reflected in the amplitude, energy, time domain. The machine learning and deep learning methods were used to verify that the pressure parameters could be predicted by the vibration sound of the heart valve. In this experiment, the authors provided a theoretical basis for the non-invasive diagnosis of pulmonary hypertension and the non-invasive measurement of pulmonary arterial pressure by simulation.

The article is interesting because the gold standard for pulmonary artery pressure is right heart catheterization (an invasive method). A non-invasive method for this assessment would be a great discovering.

The article is well written and well structured. The methods are well described. Discussions are actually the conclusions. Please make a real chapter of discussions about the importance of the results in the context of data published until now.

Answer: thank you for the comments. The authors realized that the discussions were not properly expressed. A real section of discussions was supplemented in the revised version as given in the following.

Discussions

  This study discovered the connection between valve vibration and pressure parameters through the mock-loop model, and predicted pulmonary pressure parameters based on the features of vibration waveform. Some valve vibration features discovered through the mock-loop model in this study are consistent with clinical observations in somewhat degree, as shown in Table 4. For example, both the timing of pulmonary and tricuspid valve vibration waveform reduces with respect to increasing parameters. The amplitude of vibration sound of pulmonary valve was positively correlated to pulmonary pressure.

As we all know, the pressure parameters of pulmonary circulation have important value in monitoring lung function and diagnosing pulmonary arterial hypertension. However, the pressure parameters of pulmonary circulation are not routine monitoring items in clinical practice and are invasive. The significance of the research results in this article lies in proposing a possible method to estimate pulmonary circulation blood pressure parameters using heart valve vibrations collected from the chest surface, thereby achieving continuous non-invasive monitoring. If an electronic monitoring instrument is developed based on this method, low-cost, real-time, fast, and convenient monitoring of pulmonary circulation blood pressure parameters could be achieved.

What are the limitations study? Please clearly specify.

Answer: thank you for the comment. The other reviewer also expressed similar concern. The authors specifically pointed out the limitations of this study in a section. The new section is shown in the following.

Study limitations

  The findings of this study have to be seen in light of some limitations. The human pulmonary circulatory system is very complex. Both the circuit model and the mock-loop model cannot perfectly simulate the human pulmonary circulatory system. The valves used in the mock-loop model are artificial mechanical valves, whose material and structure are largely different from those of human heart valves. The features of artificial mechanical valve vibration could be away from human heart valves. On the other hand, the changes in its hemodynamics are influenced by many factors. This study only considers a single factor: blood flow resistance caused by stenosis of distal pulmonary circulation vessels. The rules of valve vibration under multiple factors are still open to be explored.

Minor revision

There are not legends for tables.

Answer: To my understanding, you mean there are not legends for some figures. The authors have updated these figures.

There are too many figures. I suggest putting some figures in supplementary materials (figures 1,5,6, 7, 8, 9, 16, 18.)

Answer: thank you for the comment. The authors have accepted your advice. These figures are put in supplementary material for concise presentation in main text.

There are no many references probably because this study theme was not frequently searched.

Answer: The authors have searched the theme again. New more references on mock circulatory loop and applications were found and supplemented in the revised version. Thank you very much.  

[20] Shehab S, Allida SM, Newton PJ, et al. Valvular Regurgitation in a Biventricular Mock Circulatory Loop. ASAIO Journal, 2019, 65(6): 551-557.

[21] May-Newman K, Fisher B, Hara M, et al. Mitral valve regurgitation in the LVAD-Assisted heart studied in a Mock Circulatory loop. Cardiovascular Engineering and Technology, 2016, 7(2): 139-147.

[22] Tanne D, Bertrand E, Kadem L, et al. Assessment of left heart and pulmonary circulation flow dynamics by a new pulsed mock circulatory system. Experiments in Fluids, 2010, 48(4): 837- 850.

[24] Lanzarone E, Ruggeri F. Inertance estimation in Lumped-Parameter hydraulic simulator for human circulation. Journal of Biomechanical Engineering-Transactions of the ASME, 2013, 135(6).

[25] Xu KW, Gao Q, Wan M, et al. Mock circulatory loop applications for testing cardiovascular assist devices and in vitro studies. Frontiers in physiology, 2023, 14, article ID 1175919.

  By the way, the manuscript has been polished in English by an English teacher. The authors believe that the revised version is easy to read.

Round 2

Reviewer 2 Report

Thank you for responding to my comments. The mansucript was improved.